# Menstrual cycle effects on cognitive performance: A meta-analysis

**Daisung Jang**[1]*, **Jack Zhang**[2], **Hillary Anger Elfenbein**[3]

**1** Melbourne Business School, The University of Melbourne, Melbourne, Victoria, Australia, **2** Nanyang Business School, Nanyang Technological University, Nanyang, Singapore, **3** Olin Business School, Washington University in St Louis, St Louis, Missouri, United States of America

* d.jang@mbs.edu

## Abstract

Does a woman's cognitive performance change throughout her menstrual cycle? Menstruation continues to be a taboo topic, subject to myths about how it affects women. Despite the considerable number of empirical studies, there have been few quantitative summaries of what is known. To address this gap, we conducted a meta-analysis of cognitive performance across the menstrual cycle, including the domains of attention, creativity, executive functioning, intelligence, motor function, spatial ability, and verbal ability. We included studies that measured women's performance at specific points in the cycle for tasks that have objectively correct responses. Our analysis examined performance differences across phases using Hedges' *g* as the effect size metric. Across 102 articles, N = 3,943 participants, and 730 comparisons, we observe no systematic robust evidence for significant cycle shifts in performance across cognitive performance. Although two results appeared significant with respect to differences in spatial ability, they arise from a large number of statistical tests and are not supported in studies that use robust methods to determine cycle phase. Through the use of Egger's test, and examination of funnel plots, we did not observe evidence of publication bias or small-study effects. We examined speed and accuracy measures separately within each domain, and no robust differences across phases appeared for either speed or accuracy. We conclude that the body of research in this meta-analysis does not support myths that women's cognitive abilities change across the menstrual cycle. Future research should use larger sample sizes and consistent definitions of the menstrual cycle, using hormonal indicators to confirm cycle phase.

## Introduction

### Women's cognitive performance and the menstrual cycle

Women's cognitive performance throughout the menstrual cycle has been the topic of much writing and speculation. Menstruation in particular is often treated like a disease, impairing women's ability to function [1]. In 1988, the front page of the New York Times featured a story with the headline "Female sex hormone is tied to ability to perform tasks" [2]. More recently, during the 2016 US presidential election, a letter to a newspaper editor speculated

**Data availability statement:** All relevant data are within the manuscript and its Supporting Information files.

**Funding:** The author(s) received no specific funding for this work.

**Competing interests:** The authors have declared that no competing interests exist.

about candidate Hillary Rodham Clinton's ability to be president during "that time of month", despite her age strongly suggesting she had reached menopause [3]. In this meta-analysis, we examine the role of the menstrual cycle on women's cognitive performance, by conducting the most comprehensive meta-analysis on the topic to date, including in the domains of attention, creativity, executive functioning, intelligence, memory, motor function, spatial, and verbal ability, and measures of speed and accuracy in each case. Doing so allows us to address prominent limitations faced in this line of research and robustly examine the existing evidence.

## Attitudes about the menstrual cycle

Attitudes about and practices around the menstrual cycle can have detrimental impacts on women's ability to function. Menstruation is a taboo topic across many cultures, often associated with shame and misinformation [4]. Advertisements of sanitary products associate menses with themes of impediment and denial [5]. The idea of impaired functioning around menstruation has merit, based on societal resources afforded to women. Especially for those with low socioeconomic status, obtaining menstrual products can be a strain on time and resources. Obtaining menstrual products is accompanied by belittling attitudes and need for bathroom access that can be limited [6]. Such factors apply to varying degrees around the world. A systematic review and meta-analysis of menstrual knowledge and practices in India also illustrate numerous challenges for women, including being forbidden to take part in religious activities, cooking, exercise, recreation, and household activities [7]. Menstruating women also face isolation, restricted movement, and restrictions on attending social functions. In fact, only 13% of Indian women report facing no restrictions. Only a quarter of girls in India know that the uterus is the source of menstrual blood [7], which is significant because accurate knowledge about menstruation is associated with better menstrual hygiene [8]. Lack of knowledge about menstruation is prevalent across multiple cultures, particularly in low and middle income nations [9]. Such attitudes also exist in developed countries. A representative survey of over 1000 people in the US by the Tampax Corporation revealed that 35% of people believe menstruation impacts a woman's ability to think, and 26% believe women cannot function as well when menstruating [10]. Similarly, a review that examined data from university students revealed menstruation negatively impacts functioning across both low and high income countries, as women internalize menstrual stigma [8]. Attitudes and practices at both the societal and individual levels have varied across history and cultures [11,12], and can impact women's ability to function.

There are also largely negative messages communicated by individuals. Examining nearly 10,000 tweets reveals that most women sending tweets on the topic of menstruation included themes of impaired health, experience of symptoms, and stigma [13]. And although in the minority, there were also contrasting sentiments of positivity, with 14% of tweets featuring themes of strength and resilience, as well as combatting misconceptions [13], suggesting that menstruation is not always a negative experience.

In this paper we ask about the cycle's impact on women's ability to think in terms of measures that have objectively correct vs. incorrect answers. In the following sections we survey existing research on the menstrual cycle and cognitive performance, both in terms of measuring women's subjective experiences, and research that measures objective cognitive ability.

## Evidence for fluctuations in cognitive performance across the cycle

Women's reports of subjective experience would suggest that cognitive performance may fluctuate across the cycle. From the earliest attempts to measure menstrual symptoms systematically, researchers have documented significant shifts in the subjective ability to concentrate

and perform, particularly around the premenstrual and menstrual phases [14–16]. Englander-Golden and colleagues [14] note that retrospective recalls of fluctuation throughout the cycle tend to be greater than the day-to-day reports of fluctuation, which suggests that lay theory exaggerates the magnitude of self-reported cycle effects. Even with the concern about the subjective nature of self-reports, these findings suggest the possibility of shifts in the ability to focus and undertake cognitive ability tasks across the menstrual cycle. Next, we discuss cycle related shifts in brain morphology. We do so because there is a lack of research around mechanisms that directly link cognitive performance and the menstrual cycle, but changes in morphology could impact performance through secondary mechanisms, such as fluctuations in neural connectivity.

Brain imaging studies reveal shifts in morphology that complement subjective reports. Although the total volume of grey matter does not change across the cycle, localized regions show fluctuations [17,18], which correspond with subjective experiences. Fluctuations in grey matter volume in the amygdala correspond with the experience of greater negative affect from the follicular to the premenstrual phase [19]. Likewise, changes in grey matter volume in the caudate nucleus, hypothalamus, and thalamus tend to be associated with menstrual pain experience [20]. Such shifts in morphology apply to regions of the brain that are implicated in cognitive performance, which implies a potential mechanism for menstrual cycle effects on cognition. The grey matter volume in the hippocampus, a region associated with memory, increases from the premenstrual/ follicular to later phases [21,22]. This increase in hippocampus grey matter is accompanied by increased connectivity to surrounding regions [22]. When examining the entire brain, a small-scale study revealed fluctuations in patterns of atrophy across the cycle. Compared to menses, ovulation resulted in lower measured atrophy [18]. These findings are indicative of plasticity in brain structure across the cycle and suggest there could be accompanying shifts in measured cognitive ability.

Further suggestive evidence for performance differences across the cycle is evident in functional magnetic resonance imaging (fMRI) research. In this work, menstrual cycle fluctuations in estradiol and progesterone also correspond to changes in brain activity [23,24]. Activity levels of the orbital frontal cortex fluctuate with the menstrual cycle, and also correlate with changes in processing emotional stimuli [25]. In another example, increased activity between the dorsal striatum correlated with greater responsiveness to short term rewards in the follicular vs. luteal phase as demonstrated through delay-discounting tasks [26]. Moreover, multiple areas of the brain feature estrogen receptors, including the hypothalamus, amygdala, and hippocampus [27], suggesting fluctuating hormones may have an impact on their function. Taken together, these findings are consistent with the possibility of menstrual cycle effects on cognitive functioning.

## Evidence against fluctuations in cognitive ability across the cycle

In contrast with the view that the menstrual cycle influences cognitive performance, multiple narrative reviews of the existing literature on menstruation and women's cognitive performance conclude it has little impact. Noting that the discussion of the topic was dominated by informal theories and superstition as early as 1944, Seward [28] reviewed the available evidence and concluded there were little to no performance fluctuations across the cycle in both cognitive and physical tasks. Subsequent investigations reached similar conclusions [e.g., 29] as did successive reviews of the literature [10,29–32], with either null, small, or inconsistent patterns of menstrual cycle shifts in cognitive performance. Some disputed findings have reported fluctuations in cognitive ability in sexually dimorphic tasks [27,33]. The claim was that women perform better on verbal, motor, memory, and perception tasks during the luteal phase and that they perform better on visual memory, mathematic, and spatial tasks during

menses. This conclusion was challenged in a later review, which did not reveal such patterns when examining the highest quality measurements to verify menstrual phase, such as those using hormone assay or basal body temperature [34]. The more recent review included a small-scale meta-analysis of six studies that showed no improvement in mental rotation accuracy during phases that feature lower levels of estrogen and progesterone. We note that to our knowledge these narrative reviews and the small-scale meta-analysis comprise all attempts to summarize the body of research to date. The current study quantitatively analyzes cognitive performance across multiple domains of cognition, with the intended contribution to provide a comprehensive synthesis using robust methods.

To support the idea that cognitive ability does not fluctuate across the cycle, some imaging studies show no changes in ability despite changes in brain activity. For example, despite changes in susceptibility to stress and negative affect between follicular and luteal phases and the accompanying change in amygdala activity, women did not differ in reaction time in an attention task [35]. Further, despite changes in activation in the dorsolateral prefrontal cortex, which is implicated in verbal and spatial memory [36], no differences in navigation and verbal fluency were observed across menses, preovulatory, and luteal phases [23]. Similarly, fluctuations in superior and inferior parietal cortex, implicated in spatial ability, did not translate to differences in mental rotation performance across follicular and midluteal phases [24]. The lack of correspondence between physiological changes during the cycle to cognitive performance was previously observed in this literature. In a foundational study, Schmidt and colleagues [37] demonstrated that women who had high vs. low menstrual symptoms did not differ in the level of endogenous levels estrogen and progesterone.

### Evaluating the conflicting evidence amidst methodological issues

There are methodological concerns that make it challenging to evaluate the conflicting evidence in the literature. A long standing limitation in this area of research is that sample sizes are typically small, with studies sometimes reporting fewer than 10 participants [27,34]. Other challenges include inconsistent definition of cycle phases into date ranges [33,34,38,39] and reliance on self-reports of cycle phase [30] that are not reliable indicators of endocrine and other physiological changes [32]. Thus, evaluating claims about the menstrual cycle and cognitive performance would benefit from a meta-analytical approach that identifies a diverse set of studies, which is the goal of the present paper.

To address these limitations, we adopted a meta-analytic approach that encompasses major domains of cognition identified in prior research [40], including attention, creativity, executive functioning, intelligence, memory, motor function, spatial, and verbal ability. To compare cycle phases, we estimated Hedges' $g$, an effect size indicator that can account for small sample sizes. To estimate effects robustly across cycle phases, we conduct subset analyses with studies that use the authoritative definitions of the cycle established in the literature and high quality methods to determine the cycle phase, namely hormone confirmation. If a study reported multiple effect sizes, we used the composite score method to aggregate these into a single effect size [41].

### Speed and accuracy across the cycle

In our comprehensive review, we also include two different types of variables for each domain—that is, both speed and accuracy of responses across the cycle. Researchers have often used measures that contain both a timing and accuracy component. For example, the Stroop task can be measured in terms of latency of responding to stimuli or accuracy in naming stimuli. When performing such tasks, participants can make a tradeoff in how they

respond—they can take longer to deliver more accurate responses or deliver faster responses but with increased errors [42–44]. Researchers have long observed this tradeoff as an empirical regularity across a variety of domains [42,45]. Evidence suggests that speed and accuracy performance are based on accumulated information—faster responses are a product of less informed decisions whereas accuracy reflects a greater amount of information accrued [45,46]. As such, we attempt to observe general trends by analyzing response speed and accuracy scores separately.

Speed and accuracy performance depend on the type of task as well as instructions given to participants [44,47]. For example, instructions or incentives to prioritize speed or accuracy bias performance toward those goals [45]. In aggregating across multiple studies, we observed varying tasks and response formats, and due to this variability among protocols it is not possible to estimate a general speed and accuracy tradeoff function across the menstrual cycle. Nevertheless, we can still observe overall trends for providing quicker and/or more accurate responses across the cycle by separately analyzing response speed and accuracy measures.

We attempt to account for the potential effects of fatigue. In general, experimentally-induced mental and physical exertion can change speed and accuracy in dexterity tasks. Whereas mental exertion decreases both accuracy and speed [48,49], physical exertion only reduces accuracy [49]. This matters because fatigue can vary across the cycle. More than 50% of women report fatigue around the menstrual phase, even when taking oral contraceptives (OC) [50]. When mental and physical fatigue are measured separately, women report higher mental fatigue in the luteal compared to the follicular phase, while levels of physical fatigue did not fluctuate [51]. When measured with ergometers, women are most fatigued during the menstrual, followed by the follicular, then the luteal phase [52]. Such patterns suggest that with fatigue varying across the cycle speed and accuracy may vary as well.

## Overview of study

We conducted a meta-analysis of studies in which cognitive performance was measured in multiple parts of the cycle. We include studies using both longitudinal and cross-sectional designs, as well as studies that used hormonal assays or other means of cycle phase determination. This allows the possibility to observe differences in results when using varying methodologies. We report results using the guidelines recommended by the Meta-analysis of Observational Studies in Epidemiology Group [MOOSE; 53] and, where possible, the American Psychological Association's quantitative meta-analysis article reporting standards.

## Method

### Literature search strategy

We conducted a systematic review of electronic databases that could contain research about human behavior with respect to the menstrual cycle: PsycInfo, Pubmed, Social Science Research Network, Econlit, and the National Bureau of Economic Research. These databases contain behavioral research published in a journal article format, books, dissertations, and unpublished working papers. In each database, we used the following terms to retrieve documents: *menstrual, menses, estrus, oestrus, progesterone, estrogen, oestrogen, luteal, follicular*, and *ovulation*. Because we wanted to focus on the cycles of menstruating females, we set limits for databases that had this option. For Pubmed, we included articles relevant to human, female, published in English (due to resource constraints), and with participant populations aged from 6 to 64, and excluded studies that contained the terms: *embryo, cancer, treatment, pregnancy, epidemiology, infection, contract, case study, anorexia, bulimia, trial, syndrome, inflammation, congenital, disorder, pathogenesis, cell, virus, pathways, secretion, morphology,*

*substrate, immunology, implant, arthritis,* and *urinary*. For PsycInfo, we included articles that were published in English, had female participants, and with participant populations aged from 6 to 64. We placed no lower bound limitations on published date of the study, and searched for records as recent as January 2024. In addition to papers identified through this search, we also added papers from prior attempts at systematic reviews [34,54]. No specialized software was used to conduct the literature search, but the citation management software Zotero [55] was used to manage bibliographic information (e.g., remove duplicate records). All studies analyzed were from published papers or available on a dissertation database. Fig 1 shows a flowchart of the study selection process.

The first round of inclusion criteria involved examining the title and abstracts of the papers for mentions of cognitive performance across the menstrual cycle. We added to that list by conducting forward and backward citation searches of those articles using Google Scholar. We

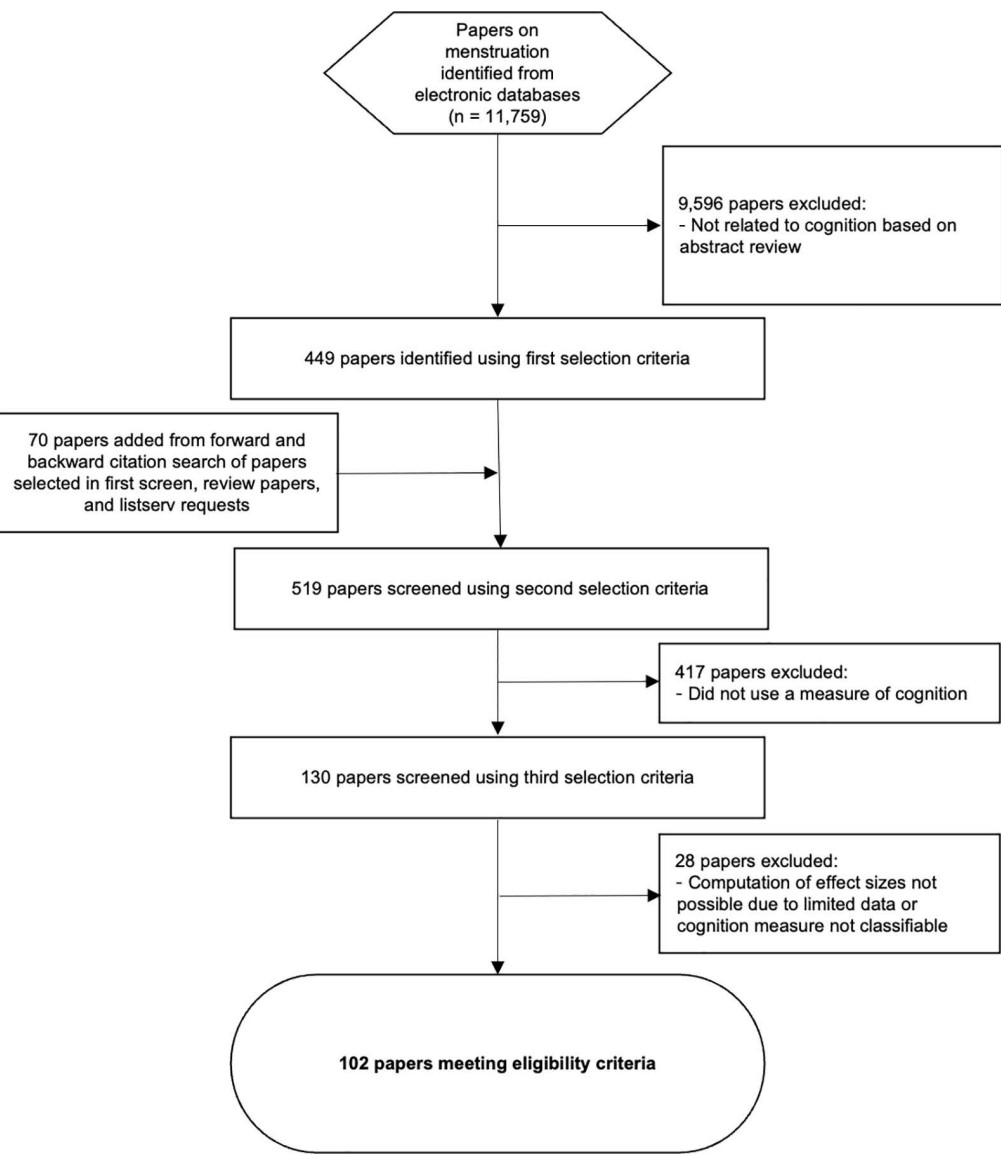

**Fig 1. Study Selection Flowchart.** Diagram of study selection process for the meta-analysis.

also requested unpublished papers and datasets on listservs for the American Psychological Association division for the Society for Experimental Psychology and Cognitive Science and the Society of Menstrual Cycle Research in February 2022. These inquiries did not result in identifying additional manuscripts or data that could be included in the analyses.

The full texts of the papers remaining in the candidate pool were subject to a second set of criteria. Papers were included if they provided the number of women measured for a cognitive ability within menstrual phases defined in terms of specific days of the cycle. We included measures of cognitive ability that were guided by prior research [40,56], in particular the domains of attention, creativity, executive function, intelligence, memory, motor function, spatial ability, and verbal ability. The full list of measures within each category appears in Table 1. Papers that did not define the menstrual cycle in terms of specific cycle date ranges and case studies were excluded because they contained insufficient information to compute effect sizes [e.g., effect size mentioned in text but could not determine the direction of effect; [57]]. In addition, we excluded samples if they measured women only at a single point in the cycle. The first author, who holds an M.A. and Ph.D. in Organizational Behavior, conducted the literature search, and extracted all data from the articles and coded all variables.

**Table 1. List of measures.**

| Domain | Measures |
|---|---|
| Attention | 2 and 7 test, attention subset Repeatable Battery for the Assessment of Neuropsychological Status, attention direction task, Brief Test of Attention, color trails I, concept formation, Continuous Performance Test, d2 test of attention, decision speed subset Woodcock Johnson III, digit span task, digit span (unspecified), digit span forwards, digit symbol, face decision task, face discrimination, figural comparison, finding As task, gestalt task, global-local task, inhibition of return task, letter detection task, lexical decision task, matching familiar figures test, number cancellation, number comparison, Paced Auditory Serial Addition Test, perceptual speed task, proof-reading test, reaction time task, serial choice task, Stroop task, target identification, trail making test A, visuospatial attention task |
| Creativity | Alternative uses task, consequences task, instances task, pattern meanings task, planning task, plot titles task, possible jobs task, similarity task, synonyms task, Torrance Tests of Creative Thinking |
| Executive function | Abstraction subset of Shipley Institute of Living Scale, arithmetical thinking, block design, color figure maze, color trails II, concept formation, Controlled Oral Word Association Test, critical thinking, deductive reasoning, digit span, backwards, figure classification, ideas generation, inference test, letter series test, mathematic test, object assembly subset Wechsler scale, phonemic fluency, rhyme generation, synonym generation, Tower of London, tracing task (complex), trail making test B, verbal fluency, word generation |
| Intelligence | Cattell Fluid Intelligence Test 20, General information subset Woodcock Johnson III, General Mental Ability Test, Multiple Choice Vocabulary test, Raven's progressive matrices, Repeatable Battery for the Assessment of Neuropsychological Status overall score, vocabulary task |
| Memory | 10/36 Visuospatial learning test, associative learning, Benton Visual Retention Test, Brief Visuospatial Memory Test Revised, California Verbal Learning Test, category exemplar generation, digit span subset Wechsler scale, digit span (forward and backward), figure memory, fragmented object identification, letter elimination, logical memory, N-back, paragraph recall, pattern recognition, picture number, picture recognition, recall task, Rey auditory verbal learning test, search test, short delay subset Wechsler scale, spatial array task, spatial recognition, spatial working memory, story recall, verbal working memory, visual memory test, visual reproduction, memory subset Wechsler scale |
| Motor function | Arm steadiness, extrapersonal motor, finger tapping, grooved pegboard, line-tracing, manual sequence box, mirror tracing, Purdue pegboard test, pursuit rotor, video game score |
| Spatial ability | Card rotations, cube comparisons, embedded figures, flags test, form board, hidden figures, hidden patterns, line orientation, mental rotation, mirror pictures, mirror tracing, navigation task, nearer point, paper folding, rod-and-Frame, size-estimation, space relations, space subset Primary Mental Abilities, spatial ability subset Woodcock Johnson III, surface development, visual matching subset Woodcock Johnson III, visuospatial subset Repeatable Battery for the Assessment of Neuropsychological Status, visuospatial composite, water level |
| Verbal ability | Anagrams, articulation, comprehension subset Woodcock Johnson III, language subset of the Repeatable Battery for the Assessment of Neuropsychological Status, object naming, verbal comprehension, verbal subset of Shipley Institute of Living Scale |

### Data coding and interrater reliability

Date of data extraction was not retained since specialized software for meta-analysis was not used. The second author served as a second coder, independently examining 15% of studies (i.e., 15) selected at random. The second coder recorded the means, standard deviations, and N associated with each dependent variable, as well as study characteristics (design, country, age mean, and age SD). There was 96% agreement between the two raters on these records. Disagreements in coded values were discussed until fully resolved.

### Definition of menstrual cycle

Apart from a nearly universal agreement across the studies sampled that a menstrual cycle is typically 28 days long, there was no consistent agreement on the number and length of phases across studies. Thus, we relied on a precedent set by [58] in their meta-analysis of pain perception across the menstrual cycle. They divided a nominal 28 day cycle into five phases: Phase 1 – menstrual (days 1-5), Phase 2 – follicular (days 6-11), Phase 3 – periovulatory, days 12-16, Phase 4 – luteal (days 17-23), Phase 5 – premenstrual (days 24-28). This division of the cycle has been used in subsequent meta-analyses [38,59]. Studies that did not use this definition to define cycle phases required further classification. If a phase definition used in the paper differed from the one above, we assigned it to the phase where the majority of its days aligned. For example, if a study defined the follicular phase as days 5-12, we categorized it under Phase 2 (the follicular phase). If a phase definition evenly straddled our defined range (e.g., luteal phase defined as days 14-19), we excluded it from analyses. To test the robustness of the results, we analyze the data using additional criteria, which we describe in a later section. Most studies did not report results by day of cycle, and so an analysis based on days of cycle as predictor was not possible.

### Computation of effect sizes

We chose Hedges' *g* as the effect size for analysis, given it is an optimal estimator for small sample sizes, which were common in the studies included. We attempted to include as many effect sizes as possible by requesting data or summary statistics from the original authors when results were verbally reported as non-significant, or when the direction of the difference could not be obtained. We obtained data for two cases where authors responded to requests (response rate = 13%). Where data or summary statistics could not be obtained, we imputed an effect size of zero for studies that reported a non-significant difference or excluded the study if we could not determine the direction of the effect size. All other studies were retained in the analyses.

Computing Hedges' *g* with repeated measures requires taking into account the test-retest correlation because ignoring reliability can result in bias [60]. Because many studies did not report test-retest reliabilities, we contacted authors of papers for the statistic or raw data and authors provided data in four cases (response rate = 9%). In total, we were able to obtain reliability for 11 studies. In the absence of primary data, we identified a meta-analysis of the test-retest reliability of cognitive measures [61] and computed the average reliability, which was.73. Where primary data were not available, we used this imputed value.

Our data contained both cross-sectional and repeated measures designs, necessitating conversion of effect sizes. Effect sizes computed for repeated measures designs and independent groups designs are not directly comparable because different degrees of freedom are used to calculate each [62,63]. Thus, we converted effect sizes from cross-sectional studies to the repeated measures metric of Hedges' *g*, imputing the average reliability observed by Calamia and colleagues [61].

Because we were interested in overall cognitive performance, we aggregated data across groups when data were reported separately for groups that had no treatment or clinical basis, such as group differences in attitudes or personality (e.g., introverts vs. extroverts; [64]), and likewise for performance differentiated by laterality (e.g., left vs. right visual field in lexical decision task; [65]). Doing so limited highly correlated effect sizes across samples. Samples served as the unit of analysis [66]. All data required to replicate the study are presented in S1 Table. We did not attempt to impute any missing effect size data. All studies included in the analyses were eligible to be included.

## Analysis strategy

We conducted a *bare-bones* meta-analysis using the *psychmeta* package (version 2.7.0) with the R statistical software language. We conducted a series of analyses comparing each phase against others, as in prior meta-analyses that examined differences across phases for effect sizes based on continuous measures [e.g., meta-analysis of pain perception across the menstrual cycle; 58]. In some cases, papers reported multiple effect sizes for the same domain of cognitive performance [e.g., reporting multiple measures of spatial ability as in 67], which creates issues of statistical non-independence. Although effect sizes could be averaged within study, the composite variable approach is favored since it generates unbiased variance estimates and an effect size of greater validity [41]. Conceptually, composite variables are analogous to deriving an average through factor analyses of multiple items vs. computing a simple average [41].

We also corrected for psychometric artifacts due to measurement errors [41,68]. Because the key test we aimed to perform was differences in performance across time, correcting for test-retest reliability was necessary to avoid issues of bias in effect sizes. As above, where test-retest reliability was not available from primary data, we imputed the meta-analytic reliability figure for cognitive measures [61]. We also excluded three extreme outliers (i.e., $g > |5|$) from analyses.

To gain deeper insight into the nature of potential differences across the cycle, we also examined separately measures of speed (e.g., response latency, time to test completion) and accuracy (e.g., percentage correct) within each domain of cognitive performance. Some papers only reported composite effect sizes of multiple measures where the unit of measurement was not specified. We included these in overall analysis while excluding them from the speed and accuracy subset analyses.

## Assessment of heterogeneity and study quality

Heterogeneity was assessed across all analyses using the $Q$ test for residual heterogeneity.

We conducted additional analyses that included only naturally cycling participants and excluded participants receiving treatments or interventions that could influence performance [e.g., participants receiving peptide injections; 69]. When examining this subset, we further excluded women with clinical diagnoses and anovulatory women. To serve as a point of comparison, we complemented these analyses with analyses of oral contraceptive (OC) users, where studies included such comparison groups. Although women on OC do not experience ovulation and therefore not a conventional menstrual cycle, they still serve as a useful comparison group against which the potential effects of hormonal shifts can be observed.

We also conducted tests to examine the robustness of results. A previous review indicated that the method of determining menstrual cycle phase was a key indication of study quality, with hormonal assays representing the highest quality [70]. Thus, we tested for the robustness of the main conclusions by comparing the pattern of results against the subset of studies that

assessed naturally cycling women whose cycles were confirmed using hormonal measures such as blood assays or luteinizing hormone tests. Other methods, such as counting days from menstruation and changes in basal body temperature were not considered in robustness tests because they are not reliable indicators of physiological changes [32]. For example, no counting method had a higher than 30% probability of determining ovulation, compared to luteinizing hormone measures [71]. In addition, we attempted to account for the varying definitions of menstrual cycles across studies. We conducted subset analyses including studies that included phase definitions that have an 80% overlap with our definition of the cycle among naturally cycling women. For example, Broverman and colleagues [72] measured participants on day 20 ± 2 days, meaning all of the days specified overlapped with our definition of the luteal phase. In contrast, Arushanyan and Borovkova [64] measured women on days 1-8, representing a 62.5% overlap with our definition of menstrual phase. While 80% is an arbitrary threshold, it provides insight into how the results might differ if cycle definitions more closely conformed to precedent definitions.

As additional analyses, we examined the cycle with respect to a three-phase cycle, defined as the menstrual phase, the ovulation phase, and an aggregate of the other phases. While we recognize that the five phase definition of the cycle is the most appropriate way to define the cycle, a three-phase definition can offer a conceptual robustness check with respect to the menstrual and ovulatory phases. This is because the menstrual phase is readily detectable via self-report and the ovulation phase is detectable through multiple physiological tests (LH surge, body temperature shift). Contrasting these phases to the remainder of phases that may be less detectable provides an additional opportunity to examine these phases with greater power. We note that this definition of the cycle for these additional analyses precludes the possibility of comparing the follicular, luteal, and premenstrual phases.

## Open practices statement

All code used for this study are available from the authors on request. No data were omitted from analyses. Data were analyzed using the R statistical programming language, using

## Results

Given the large number of results, we provide a simplified qualitative summary of them in Table 2. Key results are shown in Table 3, with detailed statistics in S2–S9 Tables.

**Table 2. Simplified Outline of Results.**

| Cognition domain | Result |
| --- | --- |
| Attention | No significant differences across the cycle |
| Creativity | No significant differences across the cycle |
| Executive function | No significant differences across the cycle |
| Intelligence | No significant differences across the cycle |
| Memory | Differences across the cycle in overall sample but none robust to multiple comparison correction or use of rigorous methods |
| Motor function | No significant differences across the cycle |
| Spatial ability | Differences across the cycle in overall sample but none robust to multiple comparison correction or use of rigorous methods |
| Verbal ability | Differences across the cycle in overall sample but none robust to multiple comparison correction or use of rigorous methods |

## Attention

When examining all samples combined, there were no significant differences in performance in attention tasks (Table 3). When examining the five-phase definition of the cycle, there were also no significant differences in speed or accuracy scores separately. Similar patterns appeared when only naturally cycling women were examined. Only three comparisons involving OC users could be tested, which were all non-significant (S2 Table). Robustness tests involving the subset of studies that overlap 80% or greater in phase definitions and studies using hormone confirmation revealed no significant differences. Heterogeneity tests were often significant when examining the full sample, but almost all were non-significant in the two robustness tests, which suggests that much of the heterogeneity can be explained by phase determination methods and disappears when examining only those studies with more stringent measures.

When examining analyses that used the three-phase definition of the cycle none of the comparisons were significant.

This is a domain of research for which there was a relatively high number of published papers (max $k = 28$).

## Creativity

When examining all samples combined, there were no differences in levels of creativity across the menstrual cycle (Table 3), when operationalizing the cycle into five phases. There were no measures of creativity based on speeded reactions, meaning that there was no opportunity to examine it in this domain. Examining only naturally cycling women also revealed no significant differences across the cycle (S3 Table). Only one study measured creativity of OC users across the cycle so we did not attempt analyses. Robustness tests with studies that have 80% overlapping phase definition and hormone confirmation studies did not reveal significant differences. Throughout, heterogeneity in the ovulatory vs. luteal phases remained unexplained.

Examining studies using the three-phase definition of the cycle also revealed no significant differences across the cycle.

The number of studies in this domain was low (max $k = 4$), suggesting much scope for additional research.

## Executive function

Executive function enables people to engage purposefully and independently in goal directed behavior [56]. When examining all samples combined and operationalizing the cycle into five phases, there were no significant differences in executive function across the menstrual cycle (Table 3). There were no differences in aggregate scores, nor when speed and accuracy scores were examined separately. This pattern was replicated among naturally cycling women and OC users (S4 Table). Robustness tests with studies that have 80% overlapping phase definitions and hormone measures for cycle determination also showed non-significant differences. Although some of the heterogeneity tests were significant when examining the full sample, the vast majority were non-significant in the robustness tests, which suggests methodological reasons for observed heterogeneity.

Examining studies using the three-phase definition of the cycle revealed no significant differences across the cycle.

The number of studies in this domain was relatively high (max $k = 21$).

## Intelligence

When examining all samples combined, there were no significant differences in apparent intelligence across the menstrual cycle (Table 3), when operationalizing the cycle into five

**Table 3. Summary of Meta-analytic Results for Entire Sample and Hormone Confirmation Studies.**

| | | Five phase comparisons | | | | | | | | | | Three phase comparisons | | |
|---|---|---|---|---|---|---|---|---|---|---|---|---|---|---|
| | | P1 vs P2 | P1 vs P3 | P1 vs P4 | P1 vs P5 | P2 vs P3 | P2 vs P4 | P2 vs P5 | P3 vs P4 | P3 vs P5 | P4 vs P5 | P1 vs. P3 | P1 vs. rest | P3 vs. rest |
| **Attention** | | | | | | | | | | | | | | |
| Entire sample | | | | | | | | | | | | | | |
| | Speed and accuracy | -0.279 | -0.024 | -0.027 | -0.038 | 0.084 | 0.282 | 0.036 | -0.085 | 0.063 | 0.315 | -0.024 | -0.083 | 0.046 |
| | Speed | -0.362 | -0.135 | -0.032 | -0.072 | 0.061 | 0.332 | 0.042 | -0.063 | 0.261 | 0.318 | -0.135 | -0.035 | 0.121 |
| | Accuracy | -0.066 | 0.122 | -0.069 | -0.217 | 0.007 | 0.031 | -0.098 | -0.230 | -0.180 | -0.246 | 0.122 | -0.144 | -0.177 |
| Hormone confirmation of cycle phase | | | | | | | | | | | | | | |
| | Speed and accuracy | -0.205 | -0.056 | 0.196 | -0.245 | -0.064 | 0.183 | -0.082 | 0.008 | 0.118 | -0.026 | -0.056 | 0.120 | 0.010 |
| | Speed | -0.470 | -0.216 | 0.080 | -0.245 | -0.074 | 0.190 | -0.030 | 0.073 | 0.264 | -0.026 | -0.216 | 0.018 | 0.055 |
| | Accuracy | -0.050 | 0.120 | 0.189 | – | -0.046 | 0.119 | -0.114 | -0.201 | – | – | 0.120 | 0.111 | -0.102 |
| **Creativity** | | | | | | | | | | | | | | |
| Entire sample | | | | | | | | | | | | | | |
| | Speed and accuracy | – | -0.817 | 0.385 | 0.089 | – | – | – | 1.528 | 0.155 | – | -0.817 | 0.569 | 0.679 |
| | Speed | – | – | – | – | – | – | – | – | – | – | – | – | – |
| | Accuracy | – | -0.817 | 0.385 | 0.089 | – | – | – | 1.528 | 0.155 | – | -0.817 | 0.569 | 0.679 |
| Hormone confirmation of cycle phase | | | | | | | | | | | | | | |
| | Speed and accuracy | – | -1.175 | 0.385 | – | – | – | – | 1.528 | – | – | -1.175 | 0.569 | 2.071 |
| | Speed | – | – | – | – | – | – | – | – | – | – | – | – | – |
| | Accuracy | – | -1.175 | 0.385 | – | – | – | – | 1.528 | – | – | -1.175 | 0.569 | 2.071 |
| **Executive function** | | | | | | | | | | | | | | |
| Entire sample | | | | | | | | | | | | | | |
| | Speed and accuracy | 0.062 | -0.022 | -0.005 | -0.023 | -0.447 | 0.079 | -0.070 | -0.056 | -0.013 | -0.017 | -0.022 | 0.041 | 0.057 |
| | Speed | – | – | -0.046 | – | – | 0.238 | 0.172 | – | – | – | – | – | – |
| | Accuracy | 0.088 | -0.041 | -0.028 | -0.023 | -0.558 | 0.078 | -0.104 | -0.101 | -0.013 | -0.017 | -0.041 | 0.046 | 0.057 |
| Hormone confirmation of cycle phase | | | | | | | | | | | | | | |
| | Speed and accuracy | – | -0.082 | 0.140 | – | – | 0.070 | -0.207 | 0.476 | – | – | -0.082 | 0.159 | 0.217 |
| | Speed | – | – | – | – | – | 0.330 | -0.134 | – | – | – | – | – | – |
| | Accuracy | – | -0.058 | 0.161 | – | – | 0.057 | – | 0.476 | – | – | -0.058 | 0.180 | 0.217 |
| **Intelligence** | | | | | | | | | | | | | | |
| Entire sample | | | | | | | | | | | | | | |
| | Speed and accuracy | – | 0.396 | -0.330 | – | – | 0.192 | – | -0.072 | – | – | 0.396 | -0.372 | -0.143 |
| | Speed | – | – | – | – | – | – | – | – | – | – | – | – | – |
| | Accuracy | – | 0.396 | -0.330 | – | – | 0.192 | – | -0.072 | – | – | 0.396 | -0.372 | -0.143 |
| Hormone confirmation of cycle phase | | | | | | | | | | | | | | |
| | Speed and accuracy | – | -0.132 | -0.162 | – | – | – | – | -0.072 | – | – | -0.132 | -0.211 | -0.143 |
| | Speed | – | – | – | – | – | – | – | – | – | – | – | – | – |
| | Accuracy | – | -0.132 | -0.162 | – | – | – | – | -0.072 | – | – | -0.132 | -0.211 | -0.143 |
| **Memory** | | | | | | | | | | | | | | |

*(Continued)*

**Table 3.** (Continued)

| | Five phase comparisons | | | | | | | | | | Three phase comparisons | | |
|---|---|---|---|---|---|---|---|---|---|---|---|---|---|
| | P1 vs P2 | P1 vs P3 | P1 vs P4 | P1 vs P5 | P2 vs P3 | P2 vs P4 | P2 vs P5 | P3 vs P4 | P3 vs P5 | P4 vs P5 | P1 vs. P3 | P1 vs. rest | P3 vs. rest |
| **Entire sample** | | | | | | | | | | | | | |
| Speed and accuracy | -0.045 | 0.045 | -0.004 | – | -0.065 | -0.023 | -0.057 | 0.167 | -0.143* | – | 0.045 | 0.001 | 0.091 |
| Speed | -0.056 | -0.070 | 0.979 | -0.227 | – | -0.099 | – | 0.691 | – | – | -0.070 | 0.331 | 1.108 |
| Accuracy | -0.040 | 0.054 | -0.084 | -0.099 | -0.087 | -0.021 | -0.106 | 0.057 | -0.175 | – | 0.054 | -0.066 | -0.022 |
| **Hormone confirmation of cycle phase** | | | | | | | | | | | | | |
| Speed and accuracy | -0.054 | -0.168 | -0.109 | – | 0.059 | -0.128 | -0.441 | 0.492 | – | – | -0.168 | -0.129 | 0.482 |
| Speed | -0.056 | -0.060 | 0.979 | – | – | – | – | 1.465 | – | – | -0.060 | 0.365 | 1.448 |
| Accuracy | -0.045 | -0.142 | -0.246 | – | 0.032 | -0.129 | -0.441 | -0.014 | – | – | -0.142 | -0.225 | -0.022 |
| **Motor function** | | | | | | | | | | | | | |
| **Entire sample** | | | | | | | | | | | | | |
| Speed and accuracy | 0.078 | 0.233 | -0.220 | – | 0.000 | -0.099 | 0.129 | – | – | – | 0.233 | -0.140 | 0.000 |
| Speed | – | – | -0.253 | – | – | 0.050 | 0.207 | – | – | – | – | -0.217 | – |
| Accuracy | 0.082 | 0.000 | -0.116 | – | 0.000 | -0.083 | – | – | – | – | 0.000 | -0.077 | 0.000 |
| **Hormone confirmation of cycle phase** | | | | | | | | | | | | | |
| Speed and accuracy | 0.247 | – | -0.286 | – | – | -0.212 | – | – | – | – | – | -0.244 | – |
| Speed | – | – | -0.510 | – | – | -0.093 | – | – | – | – | – | -0.426 | – |
| Accuracy | 0.299 | – | 0.163 | – | – | 0.115 | – | – | – | – | – | 0.057 | – |
| **Spatial ability** | | | | | | | | | | | | | |
| **Entire sample** | | | | | | | | | | | | | |
| Speed and accuracy | -0.102 | 0.098 | 0.045 | -0.125 | -0.141* | -0.179* | -0.028 | -0.017 | -0.140 | 0.108 | 0.098 | 0.038 | -0.021 |
| Speed | -0.135 | -0.116 | -0.139 | -0.227 | – | -0.261 | -0.180 | 0.200 | – | – | -0.116 | -0.131 | 0.255 |
| Accuracy | -0.119 | 0.110 | 0.072 | -0.099 | -0.141 | -0.122 | 0.068 | -0.014 | -0.140 | 0.115 | 0.110 | 0.065 | -0.020 |
| **Hormone confirmation of cycle phase** | | | | | | | | | | | | | |
| Speed and accuracy | -0.068 | 0.052 | 0.092 | – | -0.139 | -0.297* | – | 0.088 | 0.041 | – | 0.052 | 0.122 | 0.182 |
| Speed | – | -0.236 | -0.174 | – | – | -0.348 | – | 0.273 | – | – | -0.236 | -0.157 | 0.307 |
| Accuracy | 0.022 | 0.065 | 0.125 | -0.131 | -0.139 | -0.162 | – | 0.099 | 0.041 | – | 0.065 | 0.161* | 0.186 |
| **Verbal ability** | | | | | | | | | | | | | |
| **Entire sample** | | | | | | | | | | | | | |
| Speed and accuracy | – | 0.314 | 0.233 | -0.125 | – | 0.161 | -0.112 | 0.892* | -0.052 | -0.999* | 0.314 | 0.152 | 0.325 |
| Speed | – | – | 0.170 | – | – | 0.161 | – | – | – | – | – | 0.185 | – |
| Accuracy | – | 0.097 | 0.103 | -0.131 | – | – | – | 0.895 | -0.078 | -1.082 | 0.097 | 0.066 | 0.232 |
| **Hormone confirmation of cycle phase** | | | | | | | | | | | | | |
| Speed and accuracy | – | – | – | – | – | – | – | – | – | – | – | – | – |
| Speed | – | – | – | – | – | – | – | – | – | – | – | – | – |
| Accuracy | – | – | – | – | – | – | – | – | – | – | – | – | – |

All values represent Hedges' g effect sizes. P1 = Menstrual phase, P2 = Follicular phase, P3 = Ovulatory phase, P4 = Luteal phase, P5 = Premenstrual phase. Asterisks (*) indicate coefficients with confidence intervals that do not include zero. This table displays only average effect sizes for two subsets of data. Results for additional subset analyses (e.g., naturally cycling subset) and additional statistics, including including number of studies, sample size, SE, confidence intervals, credible intervals, and heterogeneity statistics are shown in S2–S9 Tables.

phases. There were no measures of intelligence based on speeded reactions in this domain. Examining only naturally cycling women also revealed no significant differences (S5 Table). No studies examined the intelligence of OC users across the cycle. Robustness tests with studies that have 80% overlapping phase definition and hormone confirmation studies also revealed non-significant differences. Although some of the heterogeneity tests were significant when examining the full sample, none were significant when examining hormone confirmation studies, which suggests that methodology was the reason for heterogeneity in the full pool of studies.

Examining studies using the three-phase definition of the cycle revealed no significant differences across the cycle.

As with creativity, the number of studies in this domain was low (max $k = 6$), suggesting much scope for additional research.

## Memory

When examining all samples combined, there were some significant differences, but these were not robust. Aggregate memory scores appeared better in the premenstrual vs. ovulation phase (Table 3), when operationalizing the cycle into five phases. However, when a Bonferroni correction was applied for multiple comparisons against the number of comparisons performed for the five-phase definition (i.e., 517 comparisons), the difference was no longer significant. Moreover, this difference was non-significant when accuracy scores were examined on their own. A similar pattern emerged amongst the subset of naturally cycling women (S6 Table). Based on the data available only one comparison was possible to examine for those who used OCs, and it revealed no significant difference between the menstrual and ovulatory phases. Robustness tests also revealed no significant differences across phases for either the 80% overlap or hormone confirmation studies. Across analyses, heterogeneity remained significant for menstrual vs. luteal, and follicular vs. premenstrual comparisons, which suggests that factors other than cycle determination methods may be involved in accounting for the variability of those effects.

Examining studies using the three-phase definition of the cycle revealed no significant differences across the cycle.

This is a domain of research for which there was a moderate number of published papers (max $k = 16$).

## Motor function

When examining all samples combined, there were no significant differences in motor function across the menstrual cycle (Table 3). Neither the overall score nor the speed and accuracy scores revealed significant differences, when operationalizing the cycle into five phases. Naturally cycling women did not exhibit differences and there were insufficient data on OC users to conduct analyses (S7 Table). Both the 80% overlap and hormone confirmation robustness tests resulted in no significant differences observed. Across analyses, none of the heterogeneity tests were significant.

Examining studies using the three-phase definition of the cycle revealed no significant differences across the cycle.

This is a domain of research for which there was a moderate number of published papers (max $k = 9$).

## Spatial ability

Some differences in spatial performance were observed with respect to the follicular phase. When examining all samples combined, there was an accuracy advantage in the ovulatory vs.

follicular phase (Table 3), when operationalizing the cycle into five phases. The same comparison was significant among the naturally cycling women (S8 Table). When Bonferroni corrections were applied, the overall sample difference was still significant. However, this comparison was non-significant in studies that measured hormones to confirm cycle differences. As such, this difference does not appear robust.

There was also a non-robust advantage for the aggregate measure of spatial performance in the luteal vs. follicular phase. This comparison was significant among naturally cycling women, and in studies that used cycle definitions that overlap 80% with ours. It was also significantly different in studies that used hormone tests to confirm phases. However, the difference was not robust to Bonferroni correction. Moreover, while the aggregate measure of spatial performance showed a robust difference, speed and accuracy measures did not independently demonstrate significance.

Tests using the three-phase definition showed some differences. Although the menstrual phase was associated with slower responses compared to the rest of the phases when comparing the overall and naturally cycling subsets, this difference was non-significant in 80% overlap and hormone confirmation subsets. And although we observed a difference between the menstrual phase and remainder of phases in the 80% overlap and hormone confirmation subsets, these differences were not robust to Bonferroni correction.

Some tests of heterogeneity were significant in the overall and naturally cycling subsample, but they were non-significant in the 80% overlap and hormone confirmation subsets. This suggests that there was heterogeneity in observed effect sizes for methodological reasons. The number of studies in this domain was relatively high (max $k = 35$).

## Verbal ability

We observed non-robust differences across the cycle. When examining all samples combined, participants had higher aggregate verbal performance in the ovulatory vs. luteal phase (Table 3). However, this was not observed in the naturally cycling subset (S9 Table), nor was the difference robust to Bonferroni correction. No data was available to examine this comparison amongst OC users, and no studies that shared 80% overlap in phase definition examined the phases. No studies in this domain examined phase differences using hormone confirmation methods.

Similarly, while there was an advantage in the premenstrual vs luteal phase based on the overall sample, which survived Bonferroni correction but evidence using more robust methods did not corroborate this pattern. The difference was non-significant among naturally cycling women, and among studies that shared 80% overlap in phase definition. As above, no studies used hormone confirmation methods in this domain.

Results using the three-phase definition revealed a non-robust difference. In the overall sample, women were more accurate in the ovulatory phase compared to the remainder of phases, but this difference could not be observed in the naturally cycling or any other subset due to the lack of available data.

Heterogeneity for the menstrual vs. luteal comparison remained significant throughout analyses, which suggests that additional factors may be involved in accounting for the variability of those effect sizes. This is a domain of research for which there was a low number of published papers (max $k = 7$).

## Examination of publication and small-study bias

We examined evidence of publication bias by examining funnel plots that placed each effect size against the standard error (Fig 2). Although outliers were present in some cases, we did

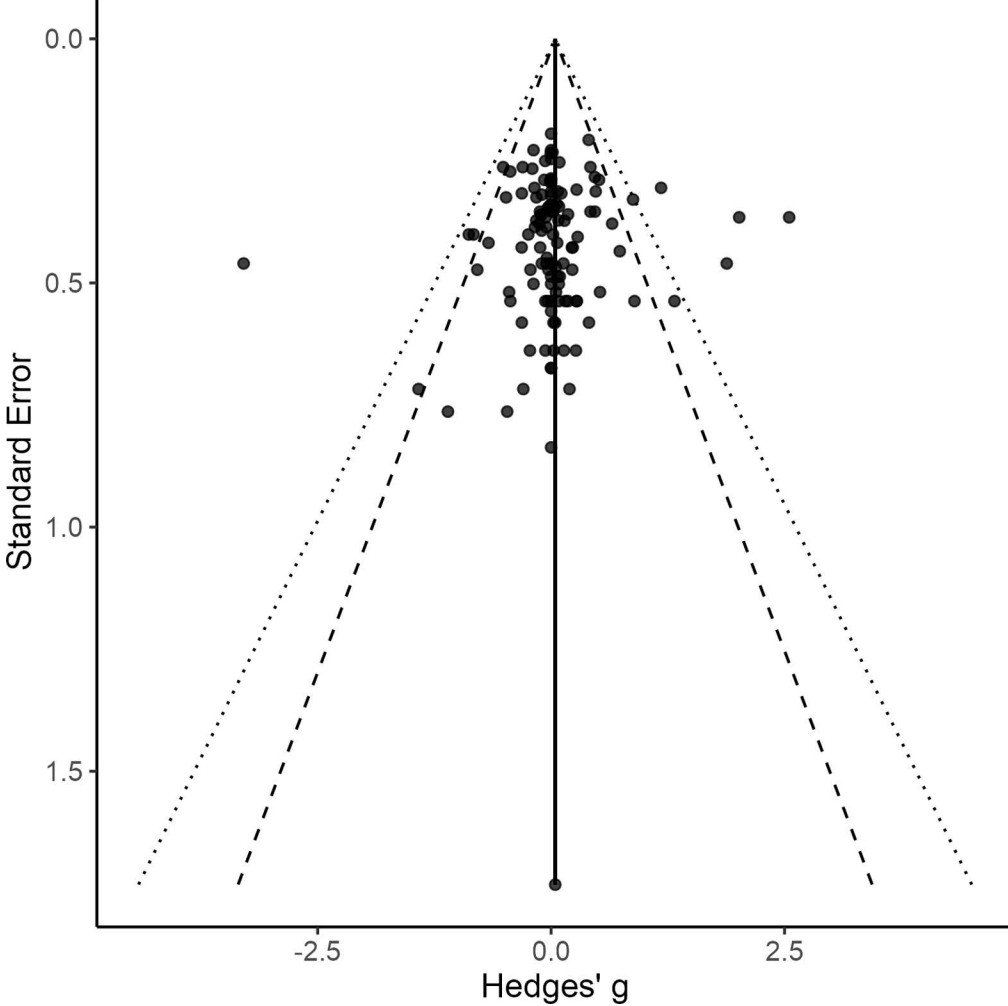

**Fig 2. Overall funnel plot.** Funnel plot of studies included in the meta-analysis.

not observe patterns of asymmetries in effect sizes that indicated publication bias. Egger's test for plot asymmetry was non-significant ($b = 0.025$, $t = -1.250$, $df = 1163$, $p = 0.212$), indicating an absence of small-study or publication bias. All articles examined in this meta-analysis are listed in S10 Table. We did not assess individual studies for risk of bias, as it was not a part of the study design.

## Power analysis

We conducted a power analysis using the metapower package (version 0.2.2) with the R statistical software language [73]. The package was developed following the power analysis approach in meta-analyses suggested by past research [74,75]. We used an estimated small-to-medium effect size of the comparisons across phases of the menstrual cycle (Hedge's *g* =.35) and the average participant sample size of the studies in our meta-analysis (N = 39) to determine the number of studies needed to detect the effects [76,77]. The analysis suggests that at least seven studies are required to achieve 80% power at an α of .05. The results caution in interpreting results for estimates involving fewer than 273 participants.

## Discussion

We find little consistent evidence that women's cognitive ability changes across the menstrual cycle when examining numerous domains of cognitive performance. Across 517 comparisons when using the five-phase definition, no difference we observed was robust to both Bonferroni corrections and supported in work that used robust methods. Similarly, across 213 comparisons when using the three-phase definition, again none were robust to Bonferroni corrections nor the use of robust methods. The lack of robust differences across a large number of comparisons suggests that there is little if any variability across the menstrual cycle in objectively scored cognitive tasks.

This lack of findings is somewhat surprising given the numerous documented physiological changes that occur across the cycle. Levels of estrogen, progesterone, and luteinizing hormone fluctuate in a well-understood manner, and receptors for estrogen in particular are present in many parts of the body, including the central nervous system and the brain [27]. Moreover, both brain structure and functional activation patterns change across the cycle [17,18,23,24], which suggests fluctuations in ability. However, the lack of evidence to support this claim suggests that the changes in brain structure and activation patterns are either small enough in magnitude that they do not have measurable influence on performance, or that women compensate for such changes through mechanisms not yet understood. Physiology does not appear to be destiny with respect to cognitive ability. Another possibility is that individual differences in responsiveness to changes in the brain are larger in magnitude than the changes themselves. This pattern was previously observed in menstrual cycle research—although levels of endogenous hormones did not differ across the cycle between those with and without premenstrual syndrome [37], individual response to hormones appeared to be responsible for premenstrual symptoms.

The results are in line with the conclusions of researchers who argued that cognitive performance does not change throughout the cycle [10,28–32,34]. Observing the small and inconsistent differences led them to conclude there was no substantive difference. Modern studies that incorporated functional brain imaging alongside cognitive measures also showed no cycle fluctuations [23,24,35,36].

This meta-analysis also examined to what extent women provide faster or more accurate responses across the cycle. Existing research on mental and physical fatigue suggested the possibility of speed and accuracy differences. However, despite the potential for greater mental and physical fatigue, particularly during the premenstrual and menstrual phases [14–16], we did not observe evidence consistent with this.

Evaluating conflicting claims about women's cognitive performance is limited by methodological challenges with menstrual cycle research. Small sample sizes and inconsistent operationalization of cycle phases make data difficult to aggregate. To account for these challenges, we conducted multiple robustness tests that evaluated evidence by examining effects among naturally cycling women, against stricter definitions of cycle phases and through hormone confirmation of phases. Doing so, only two comparisons were significant and even those resulted from a large number of unfocused comparisons and should be evaluated in that light.

In discussing these non-significant results we note that the absence of evidence for an effect is not evidence that the effect is absent. Using frequentist methods as we have done in this paper, it is not possible to conclude that a non-significant result is due to a true lack of differences between cycle phases. Some of the non-significance could be attributed to the lack of power due to small sample size, as we mention above in conducting the power analyses. A non-significant result could be due to methodological differences across studies, such as varying operationalizations of constructs (e.g., measuring attention through accuracy on the Stroop vs. color trail tasks). However we note that one prior meta-analysis that examined

a narrow set of cognitions using similar methods (mental rotation tasks) also did not find significant cycle differences [34]. Another possibility is that subgroups of women exist, who have varying levels of performance across the cycle. Aggregating across distinct subgroups may mask between group differences. A different possibility is that most of the studies were conducted on women in developed countries with a high standard of living. Such contexts should afford women with greater access to health education, hygiene products, and good nutrition. Such factors may moderate the effects of hormonal shifts. Prior evidence suggests another possibility, namely that individual differences in responsiveness to cycle shifts could be larger in magnitude compared to cycle shifts themselves [37]. Thus, caution is warranted in interpreting these non-significant results and further research is warranted.

The results we obtained have implications for policy and social justice. Women appear to think no less effectively across their menstrual cycle, even if subjective experiences can be negative in certain phases. We argue there is no scientific basis to support societal norms to eliminate or reduce women's freedoms or responsibilities through legal or organizational policy. Similarly, we argue there is no basis for attitudes, practices, and commentary that question women's cognitive performance across the cycle. Such questions leveled at candidate Hillary Rodham Clinton have undoubtedly been repeated to countless other women.

## Limitations and future directions

We find no systematic evidence of differences in cognitive performance across the menstrual cycle in the existing literature to date. That said, one can never prove a null hypothesis, but merely to establish that an effect would need to be very small if it does exist based on the methods used in existing work. There remains much room to advance knowledge in this line of research, and different types of investigations could yield different results than those reviewed here. Below we outline several areas for expanding the current body of evidence.

First, we note that some areas of research have attracted relatively little research attention and would benefit from more. For example, few studies examine verbal ability across the cycle, and none while confirming cycle phase with hormone measures. Research in the rows of our tables with missing data would serve to solidify our understanding of menstrual cycle changes in cognitive performance. We hope that this paper not only reviews what the field knows but also identifies specific areas to move the field forward.

Second, we call for more consistent operationalization of the menstrual cycle in future research. A technical difficulty we encountered was the substantial variance in how the cycle was defined in terms of a standardized 28-day cycle. This poses a challenge to accumulating knowledge. We repeat the call from prior research to standardize cycle definitions to aid future understanding [33,34,38,39,59]. In concert, we encourage the use of hormone based measures to confirm cycle phases, as counting methods can be unreliable indicators [32,71].

Third, we note the limited range of women studied in this line of research, namely eumenorrheic women in wealthy industrialized countries with relatively greater access to menstrual products and science-based education about the menstrual cycle as well as less societal and cultural stigma attached to menstruation. As we reviewed in the introduction, there are external factors that could impact women's cognition regardless of any biological differences in functioning. They involve societal, cultural, and religious beliefs that treat menstruation as a taboo [6,7,12,78]. These attitudes have been used to devalue women [79]. For example, women were judged to be of lower competence and were liked less when they accidentally dropped a tampon vs. a hair clip, suggesting cues about menstruation elicit a denigrating response [80]. This effect was similar for both women and men evaluators, suggesting that systemic attitudes can be internalized. Beyond attitudes, such attitudes toward menstruation can translate into limiting women's access to locations where they are required to perform

effectively, resulting in a lack of opportunity to demonstrate competence. These attitudes can translate to physical sanctioning, such as when menstruating women are explicitly prohibited from social activities [7]. Similarly, women of low socioeconomic status can find it difficult to access sanitary products or experience stigma when trying to access them [6]. Those products can facilitate participation, and a lack of access can lead to functional impairment. Both negative attitudes and societal exclusion can represent a persistent and substantial source of stress. Stress is known to impair cognitive function [81], so women's ability to perform cognitive tasks may be reduced by external constraints. Additional research that systematically explores external factors is necessary to understand better how to address this issue. We note that there is insufficient statistical power in our meta-analysis to explore systematically either this question or other confounding factors (e.g., level of education)—to detect a moderate effect size (.30) with 80% power in an individual study, a sample size of 84 is required, which far exceeds the average sample size of approximately 39 women.

Accumulating studies that feature larger sample sizes in diverse contexts will be necessary. Women were not included in the original studies reviewed here if they have irregular menstrual cycles or mental health diagnoses, as they are routinely excluded through selection criteria in the original studies. Better understanding of their cognitive functioning will lead to a fuller account of women's psychology overall.

Fourth, the focus of our analyses was overall cognitive functioning within a domain, and there are many subdomains with specific measures within domains, for example as conducted by Sundström Poromaa and Gingnell [34]. As such, we do not rule out the possibility that specific abilities could show cycle fluctuations even when the larger domains do not.

Fifth, although interrater reliability was high for a limited number of articles examined by two coders, the full set of articles was not coded by multiple authors. Multiple raters for each paper could increase the accuracy of the coded data.

Sixth, we could not isolate the effect of using different types of oral contraceptives (OC). Although twelve studies recruited women who use OCs, there was insufficient information provided to analyze specific types. Studies either did not report the type of OC used or reported that women used varying types of OCs. Only one study reported separate results by OC type. Investigating the association between oral contraception and cognition remains an avenue for future research.

Seventh, although we examined fluctuations in cognitive performance, we did not examine potential mechanisms that could underlie cycle fluctuations. Thus, we opted to exclude search terms such as 'morphology' in our literature search and focused on outcomes, namely measurement of performance. Future research should investigate potential mechanisms that link the cycle and cognition, to achieve greater theoretical understanding of this topic.

Finally, some of our conclusions were based on a small number of samples and/or sample size. Although our conclusions can only be based on what is published, as with any meta-analysis, smaller observations mean results are potentially affected by lack of statistical power. More research in those domains will help solidify understanding of cognitive performance in specific parts of the cycle. Likewise, additional data would be particularly worthwhile with diverse samples in terms of such factors as age, experience of pregnancy, proximity to menarche and/or perimenopause to help understand their potential influence on the association between the menstrual cycle and cognitive functioning.

## Conclusion

The topic of women's cognition and the menstrual cycle is important for a number of reasons. Studying it can enhance descriptive understanding of the cognitive functioning of half of the world's population, dispel myths, and lead to better education about women's issues. However,

this stream of literature features multiple long-standing limitations. Sample sizes are typically small [27,34], which limits generalizability of any one study. Furthermore, studies typically assess a single aspect of cognition, which limits taking a holistic perspective on the relationship between the constructs.

By conducting the most comprehensive quantitative review of the literature to date across multiple domains of cognition, we address this limitation. We observe no systematic evidence of menstrual cycle shifts in cognitive performance. This contrasts with subjective and cultural expectations, and so it is noteworthy that while using measures that have objective correct vs. incorrect responses, women show no deficits or improvements in effectiveness across the cycle. This result has implications for addressing misconceptions and myths as well as addressing discriminatory practices, as we found no scientific basis for doubting women's ability to think because of their menstrual cycle.

## Supporting information

**S1 Table. List of studies, measures, and effect sizes.** AO = anovulatory, NC = naturally cycling, OC = oral contraceptives, BBT = basal body temperature, L = longitudinal, C = cross-sectional, RBANS = Repeatable Battery for the Assessment of Neuropsychological Status, WMS = Wechsler memory scale, CVLT = California verbal learning test, BVMT-R = Brief Visuospatial Memory Test Revised, COWAT = Controlled Oral Word Association Test, WAIS = Wechsler Adult Intelligence Scale, WAIS-R = Wechsler Adult Intelligence Scale--Revised, TTCT = Torrance Tests of Creative Thinking, WJ III = Woodcock-Johnson III Tests of Cognitive Abilities. [a]Data obtained through correspondence, 7 Mar 2022. [b]Data obtained through correspondence, 7 Mar 2022. [c]Data obtained through correspondence, 17 May 2024. [d]Data obtained through correspondence, 29 Jan 2022. [e]Data obtained through correspondence, 18 Feb 2022.
(XLSX)

**S2 Table. Meta-analyses of attention measures.** All values represent Hedges' g effect sizes. P1 = Menstrual phase, P2 = Follicular phase, P3 = Ovulatory phase, P4 = Luteal phase, P5 = Premenstrual phase.
(XLSX)

**S3 Table. Meta-analyses of creativity measures.** All values represent Hedges' g effect sizes. P1 = Menstrual phase, P2 = Follicular phase, P3 = Ovulatory phase, P4 = Luteal phase, P5 = Premenstrual phase.
(XLSX)

**S4 Table. Meta-analyses of executive function measures.** All values represent Hedges' g effect sizes. P1 = Menstrual phase, P2 = Follicular phase, P3 = Ovulatory phase, P4 = Luteal phase, P5 = Premenstrual phase.
(XLSX)

**S5 Table. Meta-analyses of intelligence measures.** All values represent Hedges' g effect sizes. P1 = Menstrual phase, P2 = Follicular phase, P3 = Ovulatory phase, P4 = Luteal phase, P5 = Premenstrual phase.
(XLSX)

**S6 Table. Meta-analyses of memory measures.** All values represent Hedges' g effect sizes. P1 = Menstrual phase, P2 = Follicular phase, P3 = Ovulatory phase, P4 = Luteal phase, P5 = Premenstrual phase.
(XLSX)

**S7 Table. Meta-analyses of motor function Measures.** All values represent Hedges' g effect sizes. P1 = Menstrual phase, P2 = Follicular phase, P3 = Ovulatory phase, P4 = Luteal phase, P5 = Premenstrual phase.
(XLSX)

**S8 Table. Meta-analyses of spatial ability measures.** All values represent Hedges' g effect sizes. P1 = Menstrual phase, P2 = Follicular phase, P3 = Ovulatory phase, P4 = Luteal phase, P5 = Premenstrual phase.
(XLSX)

**S9 Table. Meta-analyses of verbal ability measures.** All values represent Hedges' g effect sizes. P1 = Menstrual phase, P2 = Follicular phase, P3 = Ovulatory phase, P4 = Luteal phase, P5 = Premenstrual phase.
(XLSX)

**S10 Table. List of articles analysed in the meta-analysis.**
(XLSX)

**S11 Table. List of articles excluded from the meta-analysis.**
(XLSX)

## Author contributions

**Conceptualization:** Daisung Jang, Hillary Anger Elfenbein.

**Data curation:** Daisung Jang.

**Formal analysis:** Daisung Jang, Jack Zhang.

**Investigation:** Daisung Jang, Jack Zhang.

**Methodology:** Daisung Jang, Jack Zhang.

**Project administration:** Daisung Jang.

**Resources:** Daisung Jang.

**Software:** Daisung Jang, Jack Zhang.

**Supervision:** Hillary Anger Elfenbein.

**Visualization:** Daisung Jang.

**Writing – original draft:** Daisung Jang.

**Writing – review & editing:** Daisung Jang, Jack Zhang, Hillary Anger Elfenbein.

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
