## [Decision Letter · Decision Letter 0]

6 Oct 2024

PONE-D-24-40134Menstrual Cycle Effects on Cognitive Performance: A Meta-Analysis.PLOS ONE

Dear Dr. Jang,

Thank you for submitting your manuscript to PLOS ONE. After careful consideration, we feel that it has merit but does not fully meet PLOS ONE’s publication criteria as it currently stands. Therefore, we invite you to submit a revised version of the manuscript that addresses the points raised during the review process.

**ACADEMIC EDITOR: **Please follow all reviewers recommendations and reply on their questions. 

We look forward to receiving your revised manuscript.

Kind regards,

Ayman A. Swelum

Academic Editor

PLOS ONE

Journal Requirements:

2. We note that you have referenced (unpublished) on page 10, which has currently not yet been accepted for publication. Please remove this from your References and amend this to state in the body of your manuscript: (ie “Bewick et al. [Unpublished]”) as detailed online in our guide for authors

3. Please respond by return e-mail with an updated version of your manuscript to amend either the abstract on the online submission form or the abstract in the manuscript so that they are identical. We can make any changes on your behalf.

4. As required by our policy on Data Availability, please ensure your manuscript or supplementary information includes the following:

Reviewers' comments:

Reviewer's Responses to Questions

**Comments to the Author**

1. Is the manuscript technically sound, and do the data support the conclusions?

Reviewer #1: Yes

Reviewer #2: Partly

Reviewer #3: Yes

Reviewer #4: Yes

2. Has the statistical analysis been performed appropriately and rigorously? 

Reviewer #1: I Don't Know

Reviewer #2: Yes

Reviewer #3: Yes

Reviewer #4: Yes

3. Have the authors made all data underlying the findings in their manuscript fully available?

Reviewer #1: Yes

Reviewer #2: Yes

Reviewer #3: Yes

Reviewer #4: No

4. Is the manuscript presented in an intelligible fashion and written in standard English?

Reviewer #1: Yes

Reviewer #2: Yes

Reviewer #3: Yes

Reviewer #4: Yes

5. Review Comments to the Author

Reviewer #1: The meta-analysis presented by the authors demonstrates a clear and well-structured analysis of the available literature. It was written clearly, with standard English and a neutral tone.

The author did not mention about the type of oral contraceptive pills. Combined oral contraceptive pills mimic the natural cycle and progesterone-only pills include progesterone and suppress oestrogen levels. Without specifying which type of oral contraceptives were used in the included studies, it becomes difficult to interpret the findings in a meaningful way, especially when evaluating outcomes related to hormonal changes. I believe it would be better to clarify which type of oral contraceptive pills were used in the meta-analysis.

Why the authors felt aggregating the follicular, luteal and premenstrual phases was required? Follicular and luteal stages are completely two different hormonal stages of the menstrual cycle. Five-phase analysis is meaningful and it correlates with the physiological changes and hormonal status of the body.

The authors have clearly outlined the methodology, and their assessment of publication bias and limitations is comprehensive. The conclusion of the study would guide the policies and social justice for women.

Overall, this meta-analysis is well-executed, making it accessible and informative for readers.

Reviewer #2: I believe that the authors of this Systematic Review and meta-Analysis put much efforts in data collection, articles selection based on the outline measures, in acompay with statistical analysis to support the manuscrpt concepts.

However, some sections (methods, discussion and limitations sections) in this manuscript require certified English review, and Proof-Reading tools use to check again at grammer level.

Regarding the "Literature search strategy" at Methods section, the authors must clarify:

- The time limit for the collected raw articles /researches (i.e., up to Month, year) at the conducted search, and emphesize if being applied differently in one or more databases.

- the authors stated that 102 articles - met criteria for inclusion - out of 11,759 potential papers being collected were selected WITHOUT specialized software to conduct the literature search, and this cannot be rational due to absolute risk of selection bias and higher rates of removing possibly included articles. Besides, a de-duplication of all articles must be applied before screening by two reviewers. Thus, there MUST be a use of a validated reference software to handle and faciliate previously essential steps for articles screening and inclusion.

- Please note that "Not assessing individual studies for risk of bias, as it was not a part of the study design" is not accepted, because non-reporting biases lead to bias due to missing evidence in a systematic review. As well, Meta-analyses are at risk of bias due to missing evidence when results of some eligible studies are unavailable because of the P value, magnitude or direction of the results. Thus, the authors must provide a list of those 102 included studies, their design and corresponding Risk of bias assessment for each by referring the modified Cochrane Collaboration tools.

Reviewer #3: Dear Author,

Thank you very much for your contribution to science. The effort needed and methodology requires a very high training and time very valuable.

Going through the document, the style, redaction and contents are easy to read and understand. The explanation of limitations and methodology are complete and adapted to the type of study. It is well addressed that some steps of a systematic review and meta-analysis are not performed (e.g. study protocol, evaluation of the risk of bias in the studies included) and such there are some limitations pointed out in the study. I would like to address some points so it is considered to add some comments on this:

- More than 50% of studies included were published more than 20 years ago. Is it any specific support to consider they are comparable to more current situations? Social, cultural, environmental issues are very likely different, and so they can affect to cognitive habilities as well.

- The range of mean ages in the studies included is really wide. Considering menstrual cycle varies accross ages and occurrence of pregnancies, proximity to menarche or to perimenpause, among other situations, this constitutes a limitation too.

As I mentioned before, I consider limitations are generally well addressed and the perspective of using this knowledge to build new and better scientific production is appropriately indicated.

Reviewer #4: Congratulations to the authors for developing a manuscript that addresses a significant research gap in the field of women’s health. The study presents a clear and well-defined research objective and draws on a substantial volume of data, which holds great potential for generating meaningful conclusions. To further elevate the quality of the manuscript, I would like to offer the following suggestions for improvement.

1. The manuscript lacks consistency in how menstrual cycle phases are defined across different studies included in the meta-analysis. Relying on self-reports for phase identification can introduce bias, especially if hormonal confirmation is absent. It may be beneficial to provide a more consistent operational definition for menstrual cycle phases across studies. If hormonal confirmation is not available for all studies, I recommend including a section on how the variability in phase definition was handled and consider conducting sensitivity analyses to assess the impact of self-reported phase data. I believe this will strengthen the reliability of the results.

2. There is no clear indication that a power calculation was performed to ensure that the meta-analysis could detect small or moderate effects across the cognitive domains examined. To increase the robustness of the findings, I suggest including a formal power analysis. This would help determine whether the study was adequately powered to detect small or moderate effects across all cognitive domains.

3. The discussion of non-significant results is somewhat lacking in depth. While null findings are reported, the manuscript could benefit from a more comprehensive discussion of the possible reasons behind these results. I recommend expanding the discussion around the non-significant findings, especially by considering methodological differences between studies, small sample sizes, or other factors that could explain the lack of significant results. Providing more context will give readers a better understanding of these findings and their implications.

4. The manuscript does not adequately address how confounding variables (such as age, education, and socioeconomic status) were controlled in the meta-analysis or the original studies. It would be meaningful to include a section discussing whether the included studies controlled for specific confounders. If this information was not available, I suggest addressing the potential impact of these factors in the limitations section to strengthen the transparency of the analysis.

5. The two statistically significant results in spatial ability may be at risk of being over-interpreted, given the large number of comparisons conducted. Given the multiple comparisons made in this study, it would be prudent to apply more stringent correction methods (e.g., Bonferroni or FDR correction) to ensure the robustness of significant findings. Additionally, a more conservative interpretation of these results may help contextualize them more appropriately.

6. The results section could be clearer. While its written comprehensively, it may be overwhelming for readers, and the key findings may be buried in the details. I recommend summarizing the key results more clearly in the main text, possibly by creating a concise summary table or figure. Detailed data can be moved to supplementary materials, allowing readers to more easily follow the central findings.

7. The manuscript touches on the cultural and societal implications of menstrual cycle research but does not explore these in enough depth, particularly regarding the influence of external factors on women’s cognitive performance. I encourage expanding the discussion of how cultural and societal factors might influence women’s cognitive performance during their menstrual cycle. Exploring these broader implications would enrich the manuscript’s relevance and offer valuable context to the findings.

6. PLOS authors have the option to publish the peer review history of their article (what does this mean? ). If published, this will include your full peer review and any attached files.

**Do you want your identity to be public for this peer review?** For information about this choice, including consent withdrawal, please see our Privacy Policy .

Reviewer #1: No

Reviewer #2: No

Reviewer #3: No

Reviewer #4: No

---

## [Author Response · Author response to Decision Letter 1]

28 Nov 2024

Dear Dr. Swelum,

We thank you for the opportunity to revise our manuscript for further consideration at PLOS ONE.

Below, we address each of the issues pointed out by the reviewers as well as required by the journal.

Yours,

Daisung Jang

Melbourne Business School

Journal Requirements:

Response: We have followed PLOS ONE’s style requirements in the revision.

2. We note that you have referenced (unpublished) on page 10, which has currently not yet been accepted for publication. Please remove this from your References and amend this to state in the body of your manuscript: (ie “Bewick et al. [Unpublished]”) as detailed online in our guide for authors

Response: On page 10, we discuss our methods for the meta-analysis, which involves potentially including results from unpublished papers. We do not cite an unpublished paper in that section or elsewhere in the manuscript.

3. Please respond by return e-mail with an updated version of your manuscript to amend either the abstract on the online submission form or the abstract in the manuscript so that they are identical. We can make any changes on your behalf.

Response: Below is the version of the abstract for both the online submission form and the manuscript:

Does a woman’s cognitive performance change throughout her menstrual cycle? Menstruation continues to be a taboo topic, subject to myths about how it affects women. Despite the considerable number of empirical studies, there have been few quantitative summaries of what is known. To address this gap, we conducted a meta-analysis of cognitive performance across the menstrual cycle, including the domains of attention, creativity, executive functioning, intelligence, motor function, spatial ability, and verbal ability. We included studies that measured women’s performance at specific points in the cycle for tasks that have objectively correct responses. Our analysis examined performance differences across phases using Hedges’ g as the effect size metric. Across 102 articles, N = 3,943 participants, and 730 comparisons, we observe no systematic robust evidence for significant cycle shifts in performance across cognitive performance. Although two results appeared significant with respect to differences in spatial ability, they arise from a large number of statistical tests and are not supported in studies that use robust methods to determine cycle phase. Through the use of Egger’s test, and examination of funnel plots, we did not observe evidence of publication bias or small-study effects. We examined speed and accuracy measures separately within each domain, and no robust differences across phases appeared for either speed or accuracy. We conclude that the body of research in this meta-analysis does not support myths that women’s cognitive abilities change across the menstrual cycle. Future research should use larger sample sizes and consistent definitions of the menstrual cycle, using hormonal indicators to confirm cycle phase.

4. As required by our policy on Data Availability, please ensure your manuscript or supplementary information includes the following:

Response: S10 Table contains a list of every paper included in the analyses. A list of every paper that was initially considered was not possible because we only retained bibliographic records of papers that passed a first relevance check for study domain. S11 Table also contains a list of papers that were excluded from the analyses that passed the relevance test but upon closer inspection did not contain sufficient data.

Response: We now better document the process by which papers were selected and updated Figure 1 to reflect this. We did not previously report the screening step prior to final analyses and do so in the revised version. S11 Table contains a list of papers that were excluded at the third selection because on closer inspection, they did not contain sufficient data, the measure of cognition was not classifiable, or the study did not operationalize the menstrual cycle in a way that could be used in the meta-analysis.

Response: All manuscripts were either published or available on a dissertation database. We note this on page 11.

Response: The first author extracted all of the data. However, because specialized software was not used to conduct meta-analyses, date of data extraction was not recorded. We now note this on page 13.

Response: We note on page 15 that all studies included in the data analyses were eligible to be included in the review.

Response: All data required to replicate the study are presented in S1 Table. We note this on page 15.

Response: Data obtained from correspondence with an author of the original research article are recorded and dates when data were obtained are recorded in the notes of S1 Table.

If applicable for your analysis, a table showing the completed risk of bias and quality/certainty assessments for each study or outcome. Please ensure this is provided for each domain or parameter assessed. For example, if you used the Cochrane risk-of-bias tool for randomized trials, provide answers to each of the signaling questions for each study. If you used GRADE to assess certainty of evidence, provide judgements about each of the quality of evidence factor. This should be provided for each outcome.

Response: On page 30, we report publication bias through the examination of a funnel plot. We also conducted a formal test of publication bias, namely Egger’s test. It showed that funnel plot asymmetry was not significant (b = 0.025, t = -1.250, df = 1163, p = 0.212), indicating there was no evidence for small-study or publication bias.

Response: We did not attempt to impute any missing data. We note this on p. 15.

Reviewers' comments:

Reviewer's Responses to Questions

5. Review Comments to the Author

Reviewer #1: The meta-analysis presented by the authors demonstrates a clear and well-structured analysis of the available literature. It was written clearly, with standard English and a neutral tone.

The author did not mention about the type of oral contraceptive pills. Combined oral contraceptive pills mimic the natural cycle and progesterone-only pills include progesterone and suppress oestrogen levels. Without specifying which type of oral contraceptives were used in the included studies, it becomes difficult to interpret the findings in a meaningful way, especially when evaluating outcomes related to hormonal changes. I believe it would be better to clarify which type of oral contraceptive pills were used in the meta-analysis.

Response: Thank you for your thoughtful comments. Unfortunately, of the twelve studies that recruited women using oral contraceptives (OC), seven did not report the type of oral contraceptive used. Of the five that did report OC types used by women, four did not report results by OC type. Within these samples, authors reported different kinds of OC being used by the recruited women. Only one study reported results by OC type. This made it impossible to meta-analyze outcomes related to specific OC formulation. We recognize this as a limitation of the study on page 37.

Why the authors felt aggregating the follicular, luteal and premenstrual phases was required? Follicular and luteal stages are completely two different hormonal stages of the menstrual cycle. Five-phase analysis is meaningful and it correlates with the physiological changes and hormonal status of the body.

Response: We thank the reviewer for this point and they are correct in that the five phase definition of the cycle appropriately separates the phases of the cycle. Thus, we focus our primary attention on analyses based on the five phase definition. The three phase definition does indeed aggregate different hormonal stages and we recognize that in the earlier version we did not provide sufficient justification for the aggregation. We chose to isolate the menstrual phase because it is readily detectable, and thus most accurately identified. We also isolated the ovulation phase because it is critical to reproduction and thus has theoretical implications. It is also detectable through multiple physiological tests (LH surge, body temperature shift). We sought to contrast these phases to the remainder of phases that may be less detectable, especially with respect to self-report or particular hormone markers. We explain this logic on pages 17-18 and appreciated the opportunity to clarify further the rationale behind the aggregation.

The authors have clearly outlined the methodology, and their assessment of publication bias and limitations is comprehensive. The conclusion of the study would guide the policies and social justice for women.

Overall, this meta-analysis is well-executed, making it accessible and informative for readers.

Response: We thank you for your helpful comments and for the opportunity to further strengthen the manuscript. Further, we appreciate your point about the value of this research for social justice.

Reviewer #2: I believe that the authors of this Systematic Review and meta-Analysis put much efforts in data collection, articles selection based on the outline measures, in acompay with statistical analysis to support the manuscript concepts.

However, some sections (methods, discussion and limitations sections) in this manuscript require certified English review, and Proof-Reading tools use to check again at grammer level.

Response: We thank you for your constructive comments and we have edited the manuscript with the assistance of a native English speaker.

Regarding the "Literature search strategy" at Methods section, the authors must clarify:

- The time limit for the collected raw articles /researches (i.e., up to Month, year) at the conducted search, and emphasize if being applied differently in one or more databases.

Response: We apologize for the misunderstanding and all databases we examined were searched for records as recent as January 2024. We clarify this on p. 11.

- the authors stated that 102 articles - met criteria for inclusion - out of 11,759 potential papers being collected were selected WITHOUT specialized software to conduct the literature search, and this cannot be rational due to absolute risk of selection bias and higher rates of removing possibly included articles. Besides, a de-duplication of all articles must be applied before screening by two reviewers. Thus, there MUST be a use of a validated reference software to handle and facilitate previously essential steps for articles screening and inclusion.

Response: We clarify our methodology further on page 11. Although we did not use specialized software to conduct a systematic search, we used citation management software (Zotero) to keep track of articles, both in terms of the originating database and in terms of managing duplicate entries. We apologize for the misunderstanding based on how we described the procedure.

- Please note that "Not assessing individual studies for risk of bias, as it was not a part of the study design" is not accepted, because non-reporting biases lead to bias due to missing evidence in a systematic review. As well, Meta-analyses are at risk of bias due to missing evidence when results of some eligible studies are unavailable because of the P value, magnitude or direction of the results. Thus, the authors must provide a list of those 102 included studies, their design and corresponding Risk of bias assessment for each by referring the modified Cochrane Collaboration tools.

Response: All articles analyzed in the meta-analysis are listed in S10 Table in the supplementary materials, and this is now mentioned on p. 30. Regarding the design of individual studies (e.g., methods used to track cycle, cross-sectional / longitudinal design / study site), these are listed on a per-sample basis in S1 Table. We analyze formally the risk of missing evidence (i.e., file drawer problem) through the examination of a funnel plot and through the use of Egger’s test for plot asymmetry (page 30), which revealed no evidence of publication bias.

Reviewer #3: Dear Author,

Thank you very much for your contribution to science. The effort needed and methodology requires a very high training and time very valuable.

Going through the document, the style, redaction and contents are easy to read and understand. The explanation of limitations and methodology are complete and adapted to the type of study. It is well addressed that some steps of a systematic review and meta-analysis are not performed (e.g. study protocol, evaluation of the risk of bias in the studies included) and such there are some limitations pointed out in the study. I would like to address some points so it is considered to add some comments on this:

Response: We thank you for your kind words and constructive comments.

- More than 50% of studies included were published more than 20 years ago. Is it any specific support to consider they are comparable to more current situations? Social, cultural, environmental issues are very likely different, and so they can affect cognitive abilities as well.

Response: We appreciate this thoughtful suggestion and in response engaged in additional analyses that examined studies conducted 20 years ago vis-a-vis those published more recently. At the end of this letter we provide a summary table analogous to Table 2 presented in the manuscript.

For studies conducted between 2004 and 2024, there were six significant differences in the overall and hormone confirmation subsets. Of those differences only one was adequately powered (i.e., sample size N = 273 or greater; see our response to reviewer 4, point 1), and involved samples that used hormone confirmation tests. However, when corrected for multiple comparisons, the p-value was no longer significant for that comparison (Phase 1 vs. remainder of phases for spatial ability accuracy).

For studies conducted in 2003 and earlier, there were more significant differences. They were observed in the domains of attention, memory, spatial ability, and verbal ability. Differences among hormone confirmation studies were observed for memory, with lower per

---

## [Decision Letter · Decision Letter 1]

22 Dec 2024

PONE-D-24-40134R1Menstrual cycle effects on cognitive performance: A meta-analysis.PLOS ONE

Dear Dr. Jang,

Thank you for submitting your manuscript to PLOS ONE. After careful consideration, we feel that it has merit but does not fully meet PLOS ONE’s publication criteria as it currently stands. Therefore, we invite you to submit a revised version of the manuscript that addresses the points raised during the review process.

**ACADEMIC EDITOR: Please respond to the following reviewer who decide against publication of your manuscript. **

We look forward to receiving your revised manuscript.

Kind regards,

Ayman A. Swelum

Academic Editor

PLOS ONE

Reviewers' comments:

Reviewer's Responses to Questions

**Comments to the Author**

1. If the authors have adequately addressed your comments raised in a previous round of review and you feel that this manuscript is now acceptable for publication, you may indicate that here to bypass the “Comments to the Author” section, enter your conflict of interest statement in the “Confidential to Editor” section, and submit your "Accept" recommendation.

Reviewer #2: All comments have been addressed

Reviewer #5: All comments have been addressed

2. Is the manuscript technically sound, and do the data support the conclusions?

Reviewer #2: Yes

Reviewer #5: No

3. Has the statistical analysis been performed appropriately and rigorously? 

Reviewer #2: Yes

Reviewer #5: No

4. Have the authors made all data underlying the findings in their manuscript fully available?

Reviewer #2: Yes

Reviewer #5: No

5. Is the manuscript presented in an intelligible fashion and written in standard English?

Reviewer #2: Yes

Reviewer #5: Yes

6. Review Comments to the Author

Reviewer #2: I think the authors have fulfilled and well-explained all peer reviewers remarks and notes on the basis of clinical and scientific tracks, and made all the recommended corrections punctually.

Reviewer #5: I have read the authors' first version of the manuscript, the comments from the reviewers, and the revised version of the manuscript. In the revised version, the authors have addressed almost all the comments and suggestions made by the reviewers in the first round of review.

Overall, I appreciate their time and effort for preparing the study. In its revised form, the article still suffers in several fronts due to incorrect use of the terms, besides the methodological issues.

A main issue about the article is that nonsignificant results may be prone to more confounds compared to significant results. The number of data points is really very small for generalizability of the findings, despite there is no barrier against repeatability. The authors do mention some of these aspects in the limitations section; however, in case of acceptance (and in case I do not miss if it was already included) they are strongly recommended to add further information that show how many data points were used for calculating the statistics in Table 3.

As for the use of the terminology, the following statement seems to be an overgeneralization of the findings obtained in the literature. For instance, under the section "Evidence against fluctuations in cognitive ability across the cycle", they state that "The claim is that on tasks where women typically outscore men (i.e., verbal, motor, memory, and perception tasks), women perform better in the luteal phase. By contrast, on tasks where men typically outscore women (i.e., visual memory, mathematical ability, and spatial ability), women purportedly perform better during menses." Given the divergent findings, the authors should avoid using the term "typically outperform".

Another term appears in the title "Beliefs about the menstrual cycle". Belief is not an appropriate term when reporting the previous research. The authors may consider using an alternative term, such as "attitude" or "social aspects of...".

"In the following sections we survey existing academic research on the menstrual cycle and cognitive performance, ...": Remove the word academic.

"The hippocampus, a region associated with memory, increases ...": What does increase exactly, please make it clear (the grey matter).

"Accumulated evidence suggests that speed and accuracy decisions are based on accumulated information": Speed and accuracy are not "decisions" but usually performance measures, used as dependent variables.

The content of the section "Evidence for fluctuations in cognitive performance across the cycle" is not compatible with its title, as it is mostly about brain physiology, not cognitive performance. The authors mention the gap between the two in later sections; however, the section needs an improvement so that it introduces the research on cognitive performance rather than the findings on brain imaging. The following section has a similar tendency to address physiological-level findings more than the findings on cognitive performance. If that is a limitation observed in the literature, the authors should emphasize that.

It is necessary to clarify why there is a need for a meta-review on the topic, despite the presence of numerous reviews. The authors mention about it in the introductory sections; however, a discussion is needed in later sections to emphasize the added value of the conducted analysis. What is the contribution of the study, different than other systematic reviews and meta-reviews on the topic?

In case of acceptance, the authors are strongly recommended on the limitations of the methodology, such as the terms used for selecting the articles and the criteria behind the selection. For instance, they used the term "morphology" as a criterion for exclusion while selecting the article from Pubmed. Nevertheless, the introduction section has a long review of morphology.

The following statements are redundant (what is the added value? If needed, they can be given as supplementary information): "The first author, who holds an M.A. and Ph.D. in Organizational Behavior, conducted the literature search.", "The first author extracted all data from the articles and coded all variables."

7. PLOS authors have the option to publish the peer review history of their article (what does this mean? ). If published, this will include your full peer review and any attached files.

**Do you want your identity to be public for this peer review?** For information about this choice, including consent withdrawal, please see our Privacy Policy .

Reviewer #2: No

Reviewer #5: No

---

## [Author Response · Author response to Decision Letter 2]

10 Jan 2025

Dear Dr. Swelum,

We thank you for the opportunity to further revise our manuscript for further consideration at PLOS ONE.

Below, we address each of the issues pointed out by the last reviewer.

Yours,

Daisung Jang

Melbourne Business School

Reviewer #5: I have read the authors' first version of the manuscript, the comments from the reviewers, and the revised version of the manuscript. In the revised version, the authors have addressed almost all the comments and suggestions made by the reviewers in the first round of review.

Overall, I appreciate their time and effort for preparing the study. In its revised form, the article still suffers in several fronts due to incorrect use of the terms, besides the methodological issues.

#1. A main issue about the article is that nonsignificant results may be prone to more confounds compared to significant results. The number of data points is really very small for generalizability of the findings, despite there is no barrier against repeatability. The authors do mention some of these aspects in the limitations section; however, in case of acceptance (and in case I do not miss if it was already included) they are strongly recommended to add further information that show how many data points were used for calculating the statistics in Table 3.

Response: We apologize for the confusion we created by limiting the results that were included in Table 3 vs. supplementary tables. Table 3 shows a summary of the full set of results. More detailed results are also presented as the reviewer requests, including number of studies, sample size, SE, confidence intervals, credible intervals, and heterogeneity statistics, which are shown in S2-S9 Tables. We now highlight this by changing the title of Table 3 to indicate that it contains summary results. In the notes section of the table, we now refer the reader to more detailed statistics in the supplementary tables. The reason for including the full statistics in supplementary tables only was to make the length of the manuscript manageable - S2 Table alone would take up 5 manuscript pages if included in the body of the text.

#2. As for the use of the terminology, the following statement seems to be an overgeneralization of the findings obtained in the literature. For instance, under the section "Evidence against fluctuations in cognitive ability across the cycle", they state that "The claim is that on tasks where women typically outscore men (i.e., verbal, motor, memory, and perception tasks), women perform better in the luteal phase. By contrast, on tasks where men typically outscore women (i.e., visual memory, mathematical ability, and spatial ability), women purportedly perform better during menses." Given the divergent findings, the authors should avoid using the term "typically outperform".

Response: We apologize for the potential overgeneralization introduced with the use of that term and have removed it from the revised manuscript.

#3. Another term appears in the title "Beliefs about the menstrual cycle". Belief is not an appropriate term when reporting the previous research. The authors may consider using an alternative term, such as "attitude" or "social aspects of...".

Response: We have replaced the word “Beliefs” with “Attitudes” throughout where appropriate.

#4. "In the following sections we survey existing academic research on the menstrual cycle and cognitive performance, ...": Remove the word academic.

Response: We have removed the word “academic”.

#5. "The hippocampus, a region associated with memory, increases ...": What does increase exactly, please make it clear (the grey matter).

Response: We have updated the section to read: “The grey matter volume in the hippocampus, a region associated with memory, increases from the premenstrual / follicular to later phases.”.

#6. "Accumulated evidence suggests that speed and accuracy decisions are based on accumulated information": Speed and accuracy are not "decisions" but usually performance measures, used as dependent variables.

Response: We have updated the section to read: "Evidence suggests that speed and accuracy performance are based on accumulated information".

#7. The content of the section "Evidence for fluctuations in cognitive performance across the cycle" is not compatible with its title, as it is mostly about brain physiology, not cognitive performance. The authors mention the gap between the two in later sections; however, the section needs an improvement so that it introduces the research on cognitive performance rather than the findings on brain imaging. The following section has a similar tendency to address physiological-level findings more than the findings on cognitive performance. If that is a limitation observed in the literature, the authors should emphasize that.

Response: We appreciate the reviewer pointing out this misalignment between the heading of the section and the content that follows. While clarifying the connection we discuss morphology to the extent that it has the potential to impact performance through secondary mechanisms. For example, changes in hippocampal volume throughout the cycle are accompanied by changes in neural connectivity (Lisofsky et al., 2015). On p. 5, we now discuss the significance of morphology before introducing research on the topic. We also note that there is a lack of research that directly addresses the case for or against shifts in cognitive performance across the cycle on p. 5.

#8. It is necessary to clarify why there is a need for a meta-review on the topic, despite the presence of numerous reviews. The authors mention about it in the introductory sections; however, a discussion is needed in later sections to emphasize the added value of the conducted analysis. What is the contribution of the study, different than other systematic reviews and meta-reviews on the topic?

Response: We thank the reviewer for the opportunity to further highlight the contribution of the manuscript and now include in the conclusion of the study (p. 37) a section that highlights the value of conducting a meta-analysis on the topic. A key methodological shortcoming of the existing literature is that most investigations utilize small sample sizes and a limited number of cognitive tasks, which makes claims about generalizability difficult. By summarizing the literature to date using a quantitative method, we are able to observe a larger picture of the role of the menstrual cycle in cognitive performance.

#9. In case of acceptance, the authors are strongly recommended on the limitations of the methodology, such as the terms used for selecting the articles and the criteria behind the selection. For instance, they used the term "morphology" as a criterion for exclusion while selecting the article from Pubmed. Nevertheless, the introduction section has a long review of morphology.

Response: As above in comment #7, we appreciate the reviewer noting the discrepancy between our discussion of morphology and demonstrated performance. We discussed morphology because it potentially contributes to performance via secondary mechanisms and now clarify why we excluded the term ‘morphology’ on p. 36—we were primarily interested in fluctuations of cognitive performance. For further responsiveness to the reviewer, on p. 36 we now discuss our methodological limitation that we only examine cognitive performance and not the underlying mechanisms of the performance. The discussion includes a suggestion that future research should identify potential mechanisms to advance theoretical understanding of this topic.

#10. The following statements are redundant (what is the added value? If needed, they can be given as supplementary information): "The first author, who holds an M.A. and Ph.D. in Organizational Behavior, conducted the literature search.", "The first author extracted all data from the articles and coded all variables."

Response: We listed the second statement in response to an editor's comment regarding the identity of the person who extracted the data. Data extraction is a separate step to the literature search in the MOOSE protocol (Stroup et al., 2000) we adhered to, hence identifying the person who conducted the search and extracted the data could potentially be different. The MOOSE protocol also requires the qualification of the person conducting the search to be listed in the manuscript. To respond to your comment regarding redundancy, we simplify the two statements to read: "The first author, who holds an M.A. and Ph.D. in Organizational Behavior, conducted the literature search, and extracted all data from the articles and coded all variables.".

References

Lisofsky, N., Mårtensson, J., Eckert, A., Lindenberger, U., Gallinat, J., & Kühn, S. (2015). Hippocampal volume and functional connectivity changes during the female menstrual cycle. NeuroImage, 118, 154–162. https://doi.org/10.1016/j.neuroimage.2015.06.012

Stroup, D. F., Berlin, J. A., Morton, S. C., Olkin, I., Williamson, G. D., Rennie, D., Moher, D., Becker, B. J., Sipe, T. A., & Thacker, S. B. (2000). Meta-analysis of observational studies in epidemiology: A proposal for reporting Meta-analysis Of Observational Studies in Epidemiology (MOOSE) group. JAMA, 283(15), 2008–2012. https://doi.org/10.1001/jama.283.15.2008

---

## [Editor Report · Decision Letter 2]

20 Jan 2025

Menstrual cycle effects on cognitive performance: A meta-analysis.

PONE-D-24-40134R2

Dear Dr. Daisung Jang,

We’re pleased to inform you that your manuscript has been judged scientifically suitable for publication and will be formally accepted for publication once it meets all outstanding technical requirements.

Kind regards,

Ayman A. Swelum

Academic Editor

PLOS ONE
---

## [Editor Report · Acceptance letter]

PONE-D-24-40134R2

PLOS ONE

Dear Dr. Jang,

I'm pleased to inform you that your manuscript has been deemed suitable for publication in PLOS ONE. Congratulations! Your manuscript is now being handed over to our production team.

Kind regards,

on behalf of

Professor Ayman A Swelum

Academic Editor

PLOS ONE